# An investigation into the effectiveness of using acoustic touch to assist people who are blind

Howe Yuan Zhu[1]*, Shayikh Nadim Hossain[2], Craig Jin[2], Avinash K. Singh[1], Minh Tran Duc Nguyen[1], Lil Deverell[3], Vincent Nguyen[1], Felicity S. Gates[3], Ibai Gorordo Fernandez[3], Marx Vergel Melencio[3], Julee-anne Renee Bell[3], Chin-Teng Lin[1]

1 University of Technology Sydney, Sydney, Australia, 2 University of Sydney, Sydney, Australia, 3 ARIA Research Pty Ltd, Sydney, Australia

* Howe.Zhu@uts.edu.au

**Data Availability Statement:** All data needed to evaluate the conclusions in the paper are present in the paper is available in a repository: https://

## Abstract

Wearable smart glasses are an emerging technology gaining popularity in the assistive technologies industry. Smart glasses aids typically leverage computer vision and other sensory information to translate the wearer's surrounding into computer-synthesized speech. In this work, we explored the potential of a new technique known as "acoustic touch" to provide a wearable spatial audio solution for assisting people who are blind in finding objects. In contrast to traditional systems, this technique uses smart glasses to sonify objects into distinct sound auditory icons when the object enters the device's field of view. We developed a wearable Foveated Audio Device to study the efficacy and usability of using acoustic touch to search, memorize, and reach items. Our evaluation study involved 14 participants, 7 blind or low-visioned and 7 blindfolded sighted (as a control group) participants. We compared the wearable device to two idealized conditions, a verbal clock face description and a sequential audio presentation through external speakers. We found that the wearable device can effectively aid the recognition and reaching of an object. We also observed that the device does not significantly increase the user's cognitive workload. These promising results suggest that acoustic touch can provide a wearable and effective method of sensory augmentation.

## 1 Introduction

Assistive technology (AT) is an extensive research field that involves designing technologies to enable an individual with a sensory disability to overcome various physical, social, infrastructural and accessibility barriers in their activities of daily living (ADL) [1]. One central area within this field is developing assistive technology for people who are blind or have low vision (BLV). According to the World Health Organisation, it is estimated that blindness affects approximately 39 million people and, additionally, 246 million people have low vision [2]. Vision can be thought of as a primary and integrated sense, and it carries an important modality. The absence of vision affects many ADLs, and the lack of access to visual information impacts one's ability to participate in a range of life and community activities [3]. Considerable research is dedicated to exploring the capability of AT [1]. With specialized Orientation and

datadryad.org/stash/share/1qWzcJGxM_
pjw3fmyTsaOOTgqqSafBUnWlPyNyzBoXl.

**Funding:** This work was supported by the
Australian Cooperative Research Centres Projects
(CRC-P) Round 11 CRCPXI000007, the ARIA
research, the University of Technology Sydney, and
the University of Sydney. Received by C.J, V.N and
C.L. Website:https://business.gov.au/grants-and-
programs/cooperative-research-centres-crc-
grants.

**Competing interests:** The authors have declared
that no competing interests exist.

Mobility (O&M) training [4, 5] to assist people who are BLV or who have low vision. The common goal for AT and specialist training for people who are BLV is to achieve equitable access to facilitate independent living, individual career choices, and provide the ability to independently perform ADL, including navigation/travel [6], hazard detection/avoidance [7, 8], locating specific household items or personal belongings [9], reading/writing [10], and engaging in social interactions (including using social media) [11]. Human performance is a complex integration of sensory and motor systems, and exploring every ADL outlined is not feasible. Therefore, this paper primarily focuses on locating specific household items or personal belongings.

Sensory augmentation is a broad AT research area, including using visual, haptic/tactile (sense of touch), and auditory feedback as a mode of augmentation; this paper focuses on using auditory sensory augmentation for people with BLV. In this case, visual sensory augmentation methods are irrelevant to our end-user group. Haptic or tactile sensory augmentation is another popular AT form for BLV users. Past works have successfully translated visual information into wearable (helmets, belts, gloves, and smart canes) haptic/tactile feedback [12–15]. The main challenge in haptic/tactile sensory augmentation is the significant training required to achieve a good spatial and semantic awareness of the user's surroundings [16]. Directional sense can only be achieved through sensory-motor coupling (relation between motor action and sensory feedback) and requires a higher level of physical and cognitive effort to gauge precise/complex spatial information. In contrast, human direction hearing is a naturally integrated sense requiring less training to perceive spatial information accurately [17–19]. Using techniques such as computer-synthesized speech (e.g., object name and direction), spatial audio, head rotation, and modulated audio (pitch and amplitude modulation), past works have found success in using auditory sensory augmentation to provide users with accurate spatial information of their surrounding environment with very little training [15, 19–22].

Many devices, both technological and mechanical in design, exist to address the various ADL needs of people who are BLV via sensory substitution and augmentation. Sensory substitution refers to converting an impaired sensory modality into another perceivable form to the user [23]. Sensory augmentation furthers this concept by using technology to augment or adapt sensory feedback to enhance the fidelity of information presented to the user [24]. The long cane is a good example of a sensory augmentation device that provides tactile sensory feedback as a mobility aid for independent travel [1]. Other more technologically-complicated examples are screen readers (text to speech), Braille displays (print to braille feedback) [25], handheld sonar devices (visual to auditory and tactile feedback) [6], and smartphone applications (Be My Eyes [26] and Seeing AI [27], visual/text to auditory feedback). The recent improvements in augmented reality (AR), wearable camera/sensor technology, and deep learning-based computer vision have led to the emergence of smart glasses as versatile and multifunctional AT [28, 29]. These systems tend to fuse a camera, Inertial Measurement Unit (IMU), microphone, Global Positioning System (GPS) and depth sensing data to provide services such as rendering environment/object/text/person as computer-synthesized speech, voice recognition control (for internet search or booking services), navigation, and voice/video communication (social call or contacting a human guide) [30, 31]. These smart glasses have many important benefits, such as being hands-free, inconspicuous in appearance, and multifunctional. Many smart glasses systems [30, 31] leverages artificial technology and speech audio feedback to deliver a reliable and robust real-time solution to aiding real-world navigation [32]. Smart glasses technology is growing in popularity and slowly being adopted within the BLV community.

Inspired by human echolocation training [4, 33], we explored the concept of "acoustic touch", coined by Jin et al. [9], which involves the use of head scanning and the activation of auditory icons (instead of computer-synthesized speech [31]) as objects appear within a

defined field of view to convey the identity and location of various items to the user. Acoustic touch builds on previous research and incorporates sensory-motor coupling via head scanning and auditory icon sonification to enhance the spatial perception of close items [9]. We propose that the technique of acoustic touch offers these three advantages:

1. **Sensory-motor coupling** to improved directional sensitivity mediated via vestibular feedback during head rotation [34–36]. As posed by Munhall et al. [37], people naturally rotate their heads to improve audio perception. By leveraging head tracking and binaural spatial audio rendering technology, acoustic touch could provide a high-quality and intuitive to localise auditory icons.

2. **Ease of integration** into conventional smart glasses technology. All commercially available AT smart glasses, such as the OrCam [38], Eyedaptic [39], and Envision [31], use integrated front-facing cameras to perform object recognition resulting in some form of 'acoustic touch'.

3. **Auditory Icons may require less cognitive processing** and be more accessible than computer-synthesized speech, especially for users with hearing loss [40]. A common practice among consumers AT smart glass is to use computer-synthesized speech audio feedback to convey information [32]. Speech is useful for conveying clear semantic information. However, speech requires a higher degree of cognitive processing and is more complex (longer time to generate and length of sound file) synthesis. In contrast, with adequate training, auditory icons could offer a more intuitive approach to presenting spatial information [41].

In this work, we compare the effectiveness of conventional speech cues to the acoustic touch technique for audio spatial information presentation with smart glasses. The paper includes a rigorous study design to evaluate the functionality and usability of the acoustic touch technique when used on a smart glasses system (our study setup in Fig 1). We hypothesize that acoustic touch can provide accurate spatial and semantic information about surrounding objects to the user without significantly increasing the cognitive demand of the user. To evaluate our hypothesis, we developed a Foveated Audio Device (FAD) to investigate the

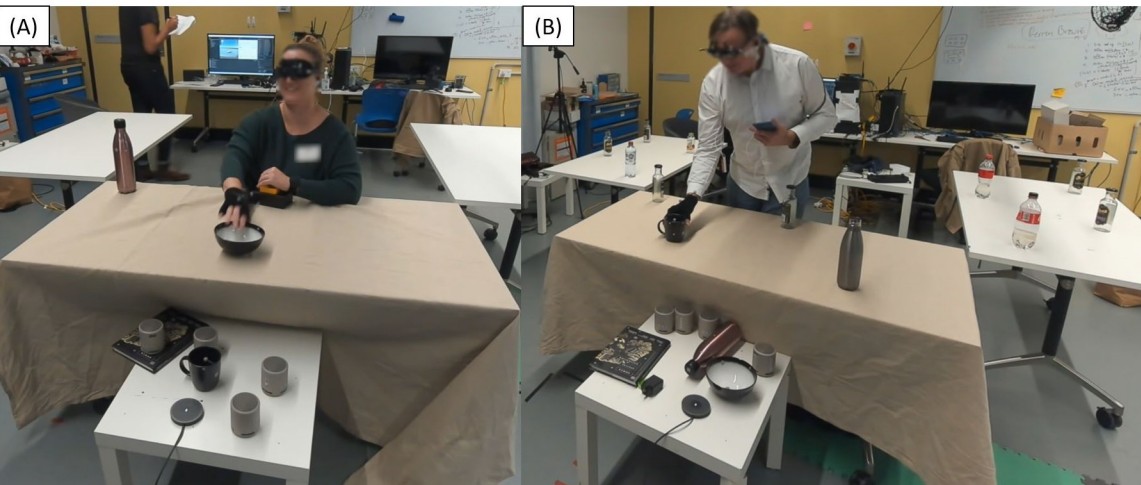

**Fig 1.** (A) The study set-up for the seated task of the study. The participants were asked to use the information provided (either using the provided wearable device or an external source) to identify and memorize the item locations on the table. The participant would then be asked to search and reach for a specific item on the table using the memorized map (without any aids). (B) The standing task requires the participant to use the provided wearable device to search and reach for a target item situated among multiple distractor items (bottles).

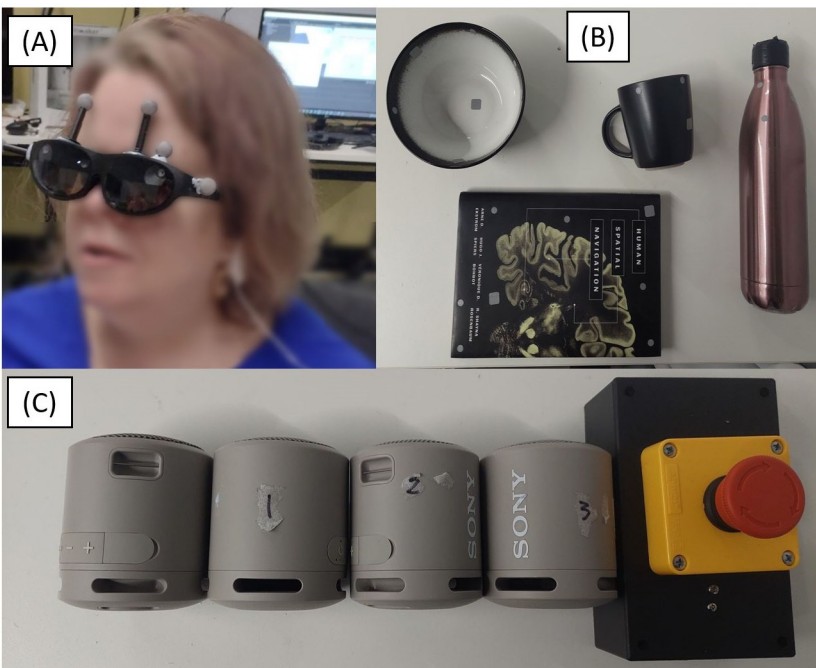

**Fig 2.** (A) The NReal AR glasses worn by a participant. The grey spheres are the reflective markers that tracked the participant's head movement through motion capture. The participant's face has been blurred for privacy. (B) The household items used during the study. The items (from left to right) are the bowl, book, cup, and bottle. (C) The Sony XB13 Bluetooth speakers and the user response button were used during the study.

use of acoustic touch in AT. The FAD utilizes AR (NReal [42], Fig 2A) glasses, computer vision, and an auditory field of view (FOV) to sonify detected items. To evaluate the FAD, we conducted a user study with 14 participants (n = 14—BLV = 7, blindfolded sighted = 7) who participated in a seated and standing reaching task, as shown in Fig 1. In both tasks, participants were presented with two or three household items (book, bottle, bowl, or cup, Fig 2B) on a table (or multiple tables) and asked to use various methods to search and reach (tap or pick up) for a single target item. The seated reaching task compares the FAD to two idealized conditions; the 'Clock' condition, which simulates an external viewer relaying item positions on a clock face (e.g., 12 o'clock bottle means the bottle is in front of the participant), and the 'Speakers' condition which indicates item location by sequentially playing auditory icons using external speakers (Fig 2C) co-located with the items. The standing reaching task primarily assesses the functional performance of the system and human behavior when using full-body movement during the search task. We functionally assessed the FAD via task performance and hand/head kinematic data recorded using a motion capture system. The usability of the FAD was assessed through questionnaires and physiological measurements, which provide insight into the participant's cognitive workload and physiological behavior (e.g., stress).

## 2 Materials and methods

### 2.1 Development of the FAD

**2.1.1 Hardware.** The FAD was developed in collaboration with ARIA Research. The FAD was co-designed with O&M specialists and multiple people with BLV (four pilot testers). We developed the FAD to explore the functionality of the acoustic touch sensory augmentation paradigm. Fig 3 demonstrates the functionality of the FAD app during the study with the

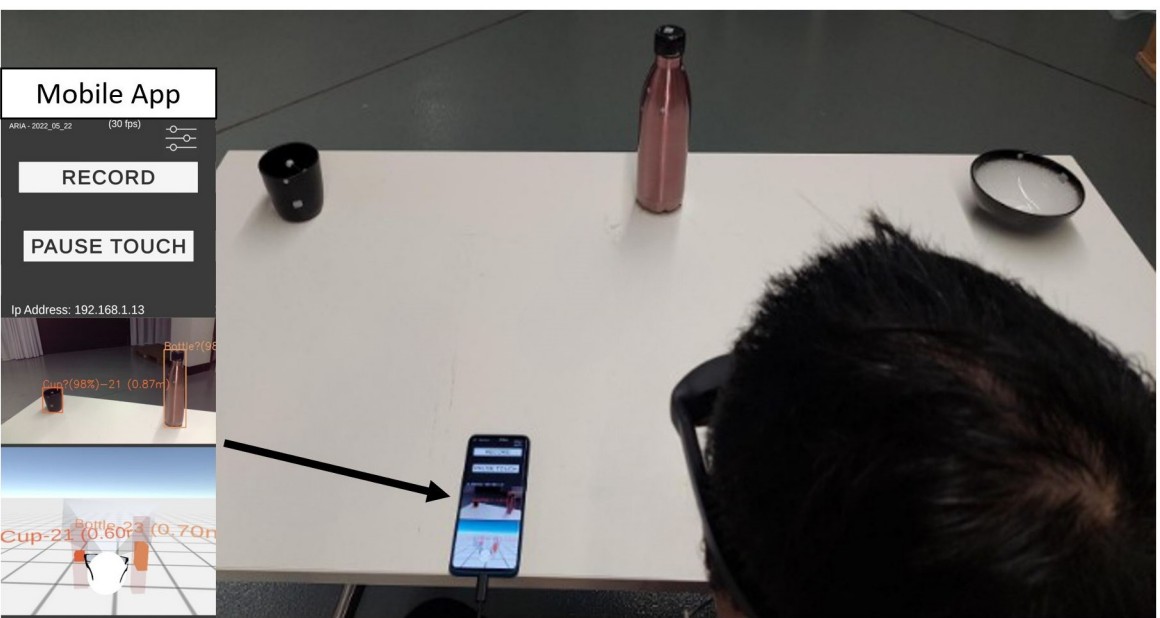

**Fig 3. A demonstration of the NReal glasses, OPPO android smartphone, and the sonification app being used to detect a cup and bottle within the user's FOV.**

mobile app view. The system consists of the NReal Light AR Glasses [42] (Fig 2A) and an OPPO Find X3 Pro Android phone [43]. The NReal glasses were chosen because of the weight (88g), computer vision support (RGB and 2x5MP cameras), 6-degree-of-freedom inertial measurement unit (IMU), binaural speakers, and compatibility with the Android Unity SDK. We attached motion capture reflective markers to the NReal smart glasses (as shown in Fig 2A) that enabled the tracking of head movement and rotations during the study. The OPPO smartphone was chosen based on processor performance and RAM size to support the deep learning object recognition model.

**2.1.2 Object recognition.** The FAD Fig 4 was developed using the Unity Game Engine 2022.1.0b2 version [44]. The Unity SDK managed the audio input and camera/head-tracking output of the NReal Glasses. Image object recognition was implemented within the Unity scene using the YOLOv5m [45] classification model. We also tested the YOLOv5s, YOLOv5l, and YOLOv5x models. but found that the m model provided the best balance of classification accuracy, inference speed, and app stability on the smartphone. The model used pre-trained weights from the COCO dataset, which contained 80 object classes [46]. For performance and app stability, the input size was set to 512x320, and the aspect ratio to 16:9. Higher resolution of the input data improved the performance but reduced the inference speed and app stability. The object recognition performed at an average of 66 ms inference speed with a power consumption of 4.5 W.

In addition to object recognition, the object's distance from the camera position is also determined (as shown in Fig 3). The distance is calculated using the stereo cameras on the glasses. We used the Semi-Global Block Matching (SGBM) [47] stereo-matching algorithm in OpenCV to calculate the depth map. Then, we used the Weighted Least Squares (WLS) filter [48] to smooth the noise in the disparity map. We aligned the depth map to the RGB frame using the cameras' calibrations. For each object, the distance was determined by calculating the average depth inside a small region (50x50) within the centre of the bounding box. Detected objects are placed into the Unity virtual environment in a position relative to the

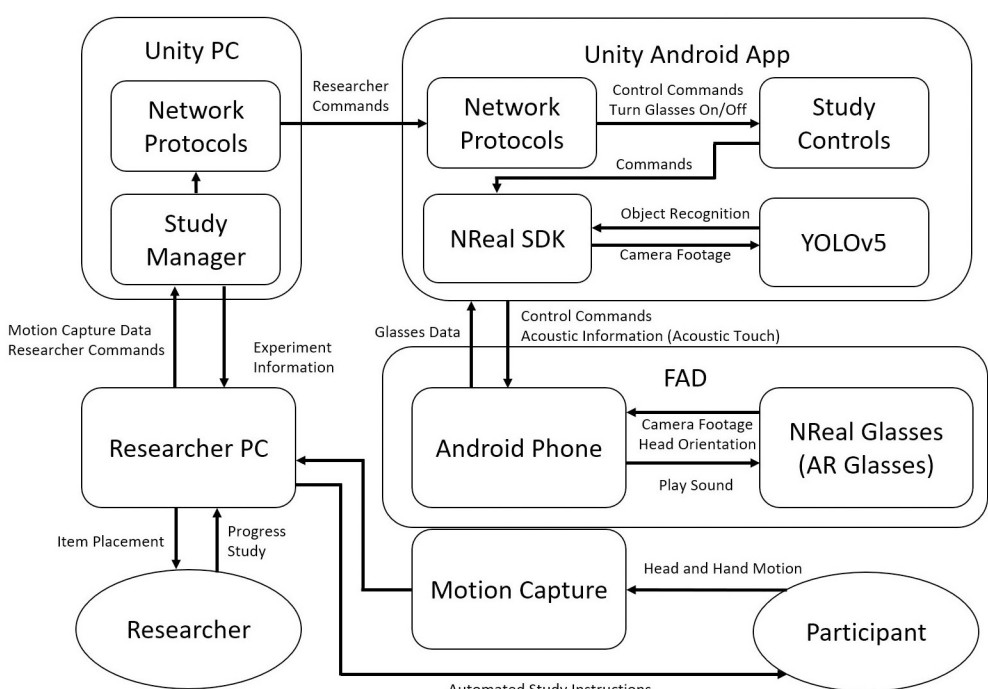

**Fig 4. A block diagram outlining the connections between the technical components and the study's users (researchers and participants).** The arrows represent the flow of information with the text specifying the type of information.

head (tracked by the IMU). For the study, we limited the object classification to four selected household items, a book, bottle, bowl, and cup. Each item was assigned an appropriate sound: the book—page turning sound, bottle—scraping a glass bottle sound, bowl—lid placed on a bowl sound, and cup—a cup placed on a wooden table sound.

**2.1.3 Auditory icon trigger parameters.** The auditory FOV and sonification parameters were achieved through a trapezoidal frustum-shaped FOV by the Unity game engine. As the user's head rotates, the repetition rate of the auditory icon sound changes based on the position of the item within the auditory FOV (the fastest repetition rate in the center of the view). During the app's development stage, we invited four pilot participants with BLV to test the FAD and the early protocols of the study. Based on the participant feedback, we selected the size of the auditory FOV, the number of FOV layers, and the sound-based repetition rate. We determined an auditory FOV size of 36.6° horizontally and 20° vertically. The FOV contained 10 layers with a 20 ms incremental delay (at the outer edge, there is a 100 ms delay between sound repetitions). This functionality improved the user's ability to precisely localize an item by fixing it to the center of their view.

## 2.2 Participants

In total, we recruited 17 participants. We evaluated a group of participants with BLV (n = 7) who served as the end users of the FAD and a group of blindfolded sighted participants (n = 7) as a control group to explore any unique differences between the groups. Three BLV participants (3 males) were excluded from the study: two withdrew due to being unable to complete the training (headache and poor hearing), and one participant had missing data due to technical issues. The study used an equal and balanced sample group of n = 14 (BLV = 7, 4 males and 3 females, and blindfolded sighted = 7, 3 males and 4 females).

The recruitment criteria for the BLV participants were: aged>18, with blindness or low vision (LogMar<2.0 visual acuity), and able to travel independently. The recruitment criteria for the blindfolded sighted participants were: aged>18 and comfortable performing tasks with a blindfold. During the screening phase, the participants provided informed consent (verbal consent from BLV participants) and were compensated for their participation. The participants provided informed consent to be recorded during the experiment and use of the footage (de-identified) for publication. The BLV participant group comprised 4 with no light perception, 2 with light perception, and 1 with low vision who wore a blindfold. The protocols and procedures of the study were approved by the University of Technology Sydney's human research ethics committee (ID: ETH22–6883). The mean age of the BLV participant group was 51.43±14.89, and the blindfolded sighted participant group was 42.43±13.56. Our blindfolded sighted participants possessed varying degrees of blindfold experience, with three being O&M specialists with experience navigating while blindfolded.

## 2.3 Study overview

Fig 4 provides an overview of the components and the connections between components within this study. To ensure consistency between participants and repeatability for future studies, the study instructions (through loudspeakers) and sequence of trials were automated through an automated manager using the Unity Game Engine on a computer. The main involvement of researchers was to set up the equipment, administer questionnaires, explain the study, and place the items on the table (using on-screen instructions, screen highlighted in Fig 5. The automated manager would communicate with the FAD using the Transmission Control Protocol/Internet Protocol commands. The commands include sending a start recording command ("RECORD" from Fig 3) at the start of the study, pausing the application ("PAUSE TOUCH" from Fig 3) between trials, and resuming at the start of the trial.

The total study session duration was around 2–3 hours. The study contained a training stage, a seated task (Fig 1A), and a standing task (Fig 1B). Participants were provided mandatory rest breaks between each stage. Questionnaires were administered during each break period. The training stage involved introducing the participant to the FAD, learning each

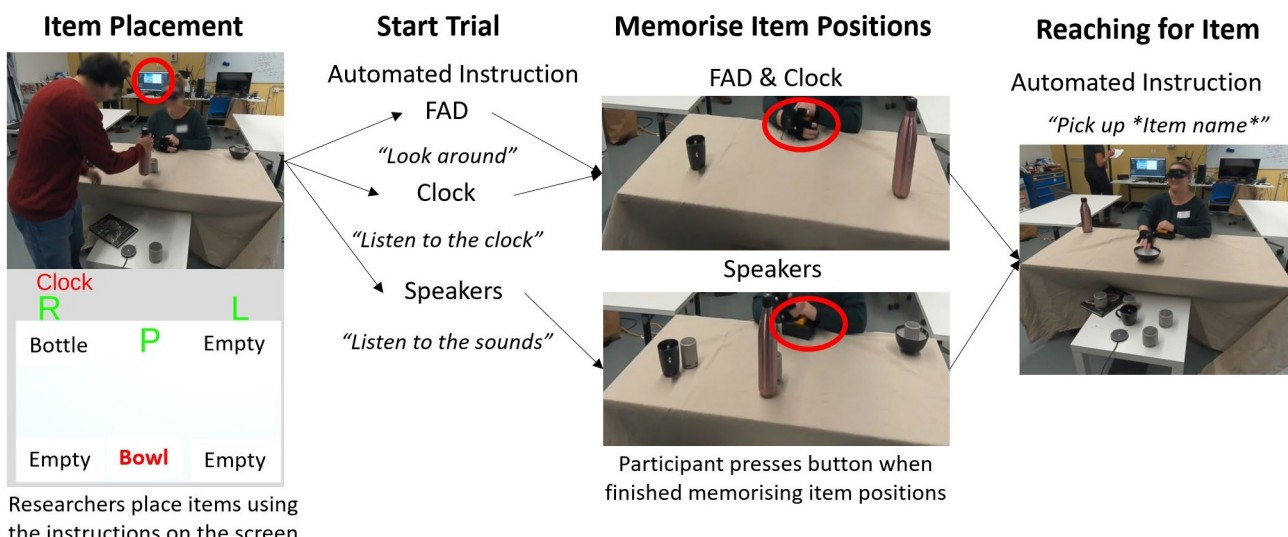

**Fig 5. An outline of the single trial protocol for the seated task.** The red circle indicates the non-automated aspect where the researcher or participant chooses to advance the trial through a key or button press.

item's sound, and practicing the experiment to ensure a basic level of proficiency. Participants were individually trained on each item, and then the number of items simultaneously placed on the table progressively increased until the participants could identify the four items on the table. The training stage finished when the participant completed 10 study trials without the researchers' assistance. We chose competency-based training conditions over a fixed training period to ensure the participants began the study tasks with adequate level of proficiency. We found that learning a new device and two other modes of presentation (clock face and external speakers) was a complex exercise, with each participant varying in the training and practicing duration. Two participants were unable to complete 10 trials successfully. Participants completed the seated (60 trials) and standing (3 trials) item-reaching task. These studies aimed to investigate the effectiveness and usability of the acoustic touch technique when used to search, identify, map, and pick up an item.

**2.3.1 Seated task.** Fig 5 outlines the flow of a single trial (three different conditions) for the seated task. The seated task was intended to compare the performance of the FAD against two idealized conditions (Clock and Speakers). This task contains the following three conditions.

- **FAD**: This condition involves using the FAD device to scan and sonify the items on the table, creating a mental spatial map of the items on the table. During the item-searching phase, the participants were asked to identify the items they heard/recognized verbally.

- **Clock**: Simulating a sighted guide or smart glasses instructing the participant on the location of each item using the spatial layout of a clock face. The instruction speaker informed the participants of the location of each item using the format of item name followed by the location on the clock ("12 o'clock Bowl"). As shown in Fig 5, from left (L, Position 1) to the right (R, Position 5): Position 1 is "9 o'clock", Position 2 is "between 10 and 11 o'clock", Position 3 is "12 o'clock", Position 4 is "between 1 and 2 o'clock", and Position 5 is "3 o'clock".

- **Speakers**: Real spatial sound is played by a loudspeaker co-located with each item on the table. This condition involves the placement of Bluetooth speakers in front of the items. These speakers will sequentially play the auditory icon three times before moving to the next item. The play sequence was ordered from left to right.

This task functionally measures three components of FAD: the ability to detect an item, recognize the sound, and memorize the position of the items. The seated task consisted of 60 trials equally distributed across three conditions (20 FAD, 20 Clock, 20 Speakers). We also varied the number of items (2 or 3) on the table for the Seated task. The task contained three conditions that were interwoven in a pseudo-randomized manner. A rest break is given after each block of 20 trials. The trial sequence was pseudo-randomized under the following rules:

- each set of 20 trials is distributed between the three conditions in a 7:7:6 distribution, with the blocks counter-balanced to ensure equal overall distribution;

- no condition appeared more than three trials in a row;

- each item was the target item (the item to be picked up) for each position (3 conditions x 4 items x 5 positions);

- each condition had 10 trials x 2 items on the table and 10 trials x 3 items on the table.

The automated manager dictates the study's flow, informs the researcher of the order of item placement, and plays instructions to the participants. The item placement stage involved the researcher placing the items on the table based on the order provided on a screen (as seen

in Fig 5. In the Start Trial stage, a speaker provided computer-synthesized oral instruction to inform the participant of the current condition of the trial. Depending on the condition, the participant would create a mental spatial map of the items on the table using the information provided. The participant then pressed a button indicating they were ready to reach the target item. The speaker then played an instruction to pick up a specific target item the participant would reach. No auditory stimuli were provided during the reaching stage; the participant must rely on their spatial memory alone to reach the object. The participants were instructed only to use their right hand and were allowed one reach attempt per trial. Minor adjustments to their hand trajectory were allowed. The reach was considered successful if the participant touched the item and unsuccessful if their arm was extended incorrectly.

**2.3.2 Standing task.** The standing task is a functional test that explored the performance of the FAD when searching for an item with full mobility and within a cluttered environment. As shown in Fig 1B, the standing task involved three tables placed around the standing participant in a U shape. Twelve differently shaped and sized bottles are placed around the participant, with four bottles per table. The participant was asked to search and pick up a non-bottle item among multiple bottles, which served as distractors. This task consists of 3 trials which involved reaching for the book, bowl, or cup. In each trial, the researcher replaced one bottle with the target item according to the placement position on the screen. Then, the automated manager instructed the participant to "look for the" target item (e.g., "look for the bowl"), prompting the participant to search and reach it. Participants were given unlimited reach attempts but were allowed to give up if they could not find the item.

## 2.4 Study items and equipment

Fig 2B presents the four household items chosen for this study. From left to right, there is the bowl, book (titled "Human Spatial Navigation"), cup, and bottle. These items have reflective stickers attached for position tracking using the motion capture system. A foam pad was placed on the bottom of each item to reduce noise and auditory cues during item placement. The tablecloth also assisted with reducing the item placement noise.

Fig 2C shows the four Sony XB13 Bluetooth speakers [49] and the user response button used to signal the completion of the memorizing stage. The unlabelled speaker is the instruction speaker used to relay all automated instructions. The speakers labeled with a number were used for the Speakers condition to sequentially play item sounds from their position co-located with items on the table. The speakers were always placed from left to right, with 1 being the leftmost item position. An RME Fireface 400 sound card was used to interface the automated manager and the Bluetooth Speakers. The user response button was held under the participant's left hand (the right hand was used for reaching).

## 2.5 Measurement metrics and apparatus

**2.5.1 Task performance.** For the seated task, we assessed the participant's ability to generate and memorize a spatial map (when exploring the items on the table) and the ability to use the spatial map to locate a specific item. We assessed the participant's ability to find a specific item in a cluttered environment for the standing task. The seated and standing task performance is measured using an annotated trial run sheet of events during the trials. For every trial, the observing researcher noted the trial number, trial condition, the outcome (success or failure) of contacting the target item, the recognized items (spoken verbally during the FAD condition), and any other noteworthy events. After the session, the trial run sheet was manually digitized and processed for each participant. The participant data were separated into groups (BLV and blindfolded sighted).

The task performance results were calculated using the following equation:

$$Task\ Success\ Rate = 100 \times \frac{Number\ of\ Successful\ Trials}{Total\ Number\ of\ Trials}$$

$$Item\ Detection\ Rate = 100 \times \frac{Number\ of\ Successful\ Detections\ of\ the\ Item}{Total\ Number\ of\ Occurances\ of\ the\ Item}$$

The time required for the participant to build a mental spatial map was recorded through an event log with timestamps. The exact time was calculated by comparing the timestamp between the start of the trial and the timestamp when the participant pressed the user response button. It would not be meaningful to compare the conditions as the Clock and Speakers conditions required no active tasks while the FAD condition required active scanning, which naturally took longer. We mainly used this data point to explore the differences between BLV and blindfolded sighted participant groups.

**2.5.2 Motion capture body tracking.** Twelve Optitrack Flex13 [50] cameras were used as the optical motion capture to track the item positions, participants' right-hand movement, and their head/FAD rotation. As shown in Fig 6A, the motion capture system used reflective markers to translate physical items/body parts to rigid bodies, which were converted to trackable game objects. The motion capture data for the hand, head, and items were saved to a local JSON file through the Unity Game Engine. For the seated task, we measured the hand and head motion capture data to track the functional behaviors of the participants. We did not use

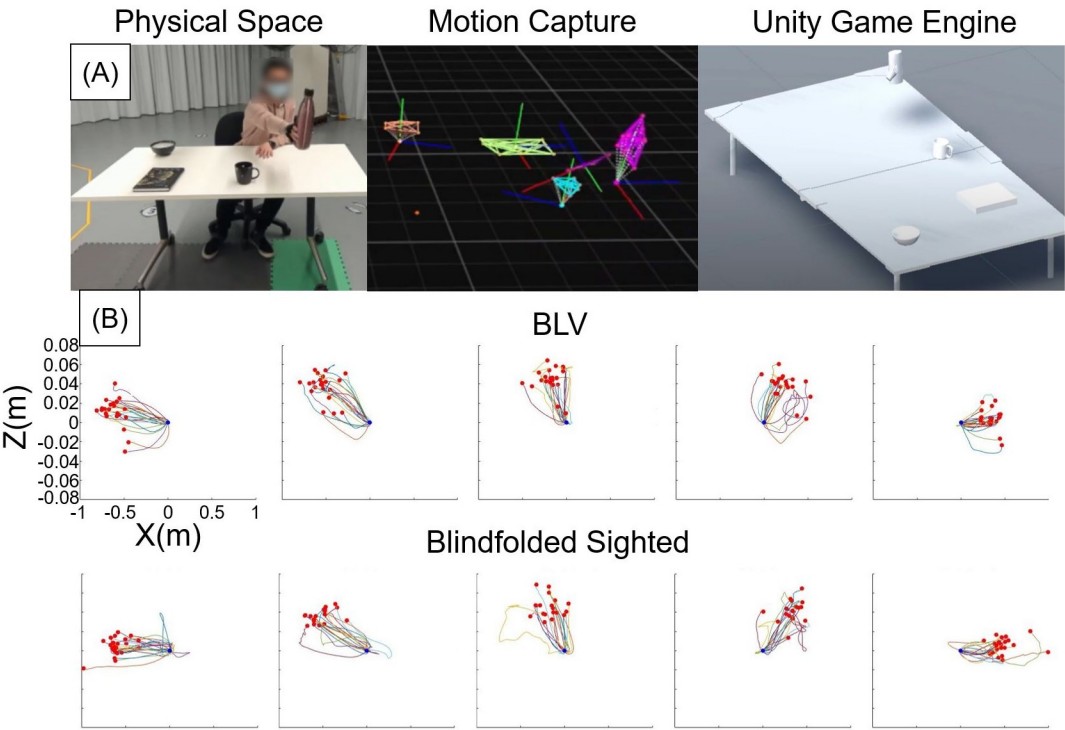

**Fig 6.** (A) depicts the hand, head, and item tracking through motion capture in this study. (B) Plots of the hand trajectory (z-x plane, overhead view) for every FAD trial by the participants. The blue dots represent the hand's starting position with an offset always initialized at (0,0). The red dots represent reached item position. Position 1–5 refers to reaching positions from the left of the user to the right.

the motion capture system for the standing condition because of limitations in the tracking area size.

We collected the hand kinematic data to evaluate the reaching behavior of the participants. The trajectories were sorted by the z-x plane (overhead view) with an offset applied so that all the trajectories started from (0,0). We observed the inconsistent ending points of the data (see Fig 6B). This inconsistency occurred due to the inconsistency in the starting position of the hand and the ending position of the item placement. We explored using techniques such as dynamic time warping [51] to normalize trajectories to the same length, scale, and start/end points. However, the resulting data became too distorted to attain any meaningful information. Instead, we calculated the hand-reaching time, reaching speed, and trajectory-to-optimal-path ratio. The Hand-reaching time is the time between the participant receiving the instruction and the participant's hand touching the item. The Hand reaching speed was calculated by dividing the total trajectory length by the reaching time. The trajectory-to-optimal ratio refers to the percentage difference between the trajectory the participant takes and the optimal (direct) route from the start to the endpoint. The trajectory-to-optimal path ratio was calculated using the following equation:

$$Trajectory\ to\ Optimal\ ratio = 100 \times \frac{total\ trajectory\ length}{distance\ between\ start\ and\ end\ point} - 1$$

The head kinematic data provides insight into the search patterns and the efficiency of the FAD for identifying items. We only used the FAD condition data for head rotation because the head movement was not required for the Clock and Speakers conditions during the spatial mapping stage. We sorted the head rotation into a single-axis orientation (y-axis) and calculated the values for both successful and unsuccessful (Missed) reaching trials. We calculated the total head rotation sorted by groups and the rotation speed (total head rotation/user response time) sorted by groups. This measurement helps understand the relationship between head rotation and the success of a task. One limitation of this measure was the imbalance between the successful and missed trials. Overall, most of the trials were successful (85.53 ±14.93%) in reaching the target item; fewer missed trials influenced the reliability of the date for the missed trials.

**2.5.3 NASA-TLX.** We investigated the participant's cognitive state to explore whether the FAD increased the participant's mental workload levels. We used the NASA Task Load Index (NASA-TLX) questionnaire to gauge the participant's mental workload. The NASA-TLX survey provided a reliable, self-reported metric to assess an individual's subjective mental workload [52]. We assessed the seven aspects of mental, physical, temporal (feeling rushed/hurried), performance, effort, frustration, and the total score (RAW-TLX). The participants responded to the questionnaire in three stages after the Training session: the Seated and Standing tasks. We used the unweighted 7-point (scaled to 21 points) NASA-TLX questionnaire [52]. We averaged scores across groups and compared the mental, physical, temporal, performance, effort, frustration, and total (RAW-TLX) metrics over the three stages. This questionnaire indicates the participant's level of perceived workload throughout the study.

**2.5.4 Physiological measurements.** The participant's physiological state plays a vital role in understanding the usability of the FAD device. The FAD device should not unintentionally increase the participant's stress level or induce anxiety. We suspect factors such as uncertainty of the device performance and frustration/discomfort when using the device could increase the participants' stress/anxiety levels. We used two physiological measurement devices in this study. Firstly, we used the Zephyr bioharness module [53] to measure the participant's Heart Rate (HR), Heart Rate Variability (HRV), Breathing Rate (BR), and Skin Temperature

(TEMP). The HR and HRV data were filtered through the HR confidence (remove when HR Confidence<20%) metric provided by Zephyr [54]. The BR and TEMP were filtered for outliers using two standard deviations from the median as the boundary condition. The second device was the Empatica E4 wristband [55], which measured the participant's Electrodermal Activity (EDA) response. The EDA data were filtered for outliers using two standard deviations from the median as the boundary condition. The physiological measures were processed and averaged across the participant groups. We compared the physiological measures by condition to observe any specific condition that increased activity. We also compared the Seated and Standing tasks to observe for changes in physiological activity. These measurements provided helpful information regarding any increase in stress levels.

## 2.6 Data analysis and statistics

The measurements, numerical results, and figures were processed using Matlab. For the statistical analysis, we determined the normality of each measure by applying the one-sample Kolmogorov-Smirnov test, which compares the empirical data to a standard cumulative distribution function. Datasets for both the BLV and Sighted groups were tested with the significance level criterion of $\alpha < 0.05$. We found the various datasets were of a non-normal distribution of data (both groups p<0.001). We also confirmed the results visually through a cumulative distribution function (empirical vs normal) plot. Therefore, for inter-group (BLV vs blindfolded sighted) comparison, we applied a Wilcoxon rank sum test (or Mann-Whitney U-test) to compare in a pairwise manner. For intra-group (within-group) comparison, we applied a Wilcoxon signed-rank test. The significance level ($\alpha$) of 0.05 is set to determine statistical significance. The significance stars for the figures are *p<0.05, **p<0.01, and ***p<0.001.

## 3 Results

### 3.1 Seated task performance

Fig 7A presents the success rate of the participants picking up the correct item. We found that the BLV participant group's performance (FAD 81.66±13.72%, Clock 90.71±11.70%, and Speakers 85±13.54%) when using the FAD was comparable to the idealized conditions. In contrast, we observed that the blindfolded sighted participant group performed worse when using the FAD (FAD 73.01±18.92%, Clock 93.78±7.39%, and Speakers 89.03±16.66%). The blindfolded sighted participants were significantly lower in success rate in the FAD condition compared to the Clock (Z = 2.21, p = 0.027, and r = 0.83) and Speaker condition (Z = 2.21, p = 0.027, and r = 0.83). Interestingly, we did not observe any performance differences between the O&M specialists (n = 3, 64.80±26.15%) to the non-specialist blindfolded sighted participants (n = 5, 79.17±11.90%).

Fig 7B presents the success rate of the participants in detecting the items on the table during the searching phase of the FAD condition. During the FAD condition, the participants orally identified any item sounds they heard during the searching phase. This indicated the FAD's effectiveness in accurately conveying information to the user. We found that the BLV participant group could detect each item with a similar level of success. For the blindfolded sighted participant group, we observed a decrease in performance (-12.14%) when detecting the bowl compared to the cup (Z = 2.20, p = 0.028, and r = 0.83). Through observations and the trial sheet notes, 13 out of the 14 participants reported confusion between the bottle, bowl, or cup sound, leading to a few unsuccessful trials. This confusion between items was also observed with the blindfolded sighted participant group performing significantly better in the Clock condition (speech) for the bottle and bowl when compared to the FAD (auditory icons)

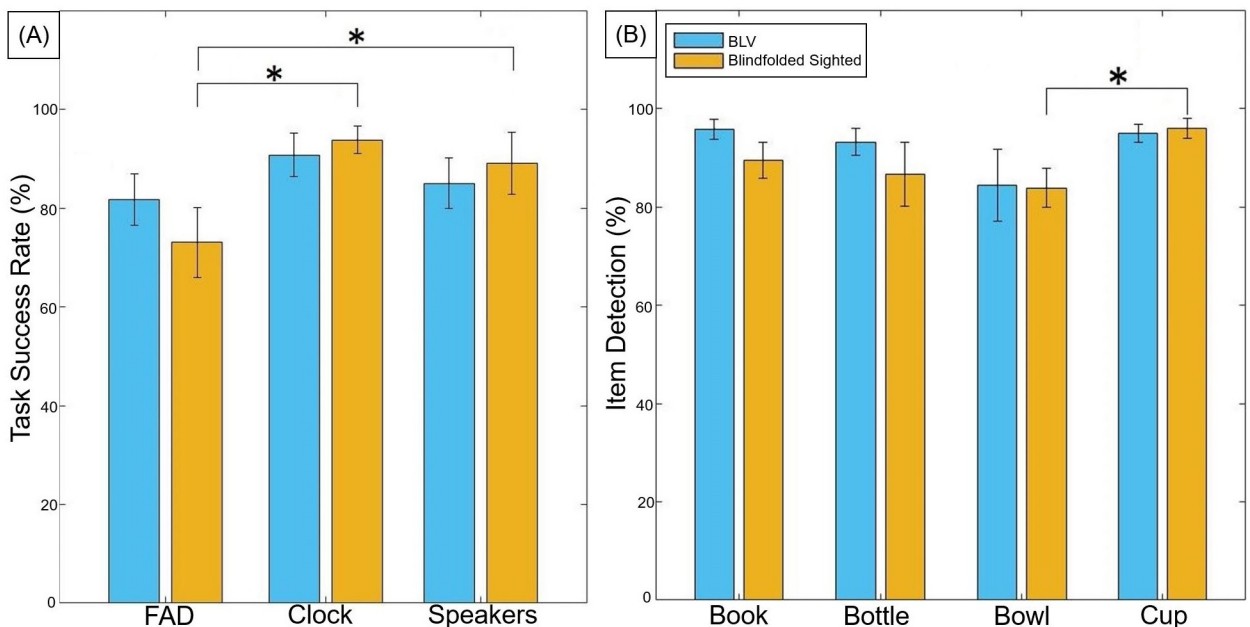

**Fig 7.** (A) The average with standard error bars of the seated task success rate sorted by each condition for both the BLV and blindfolded sighted participants. (B) The average with standard error bars of each item's detection rate when using the FAD to search for the items on the table.

(Z = 2.20, p = 0.028, and r = 0.83, Fig 8A). This was likely due to the closeness of the bowl, bottle, and cup sounds being all glassware. We found that both participant groups took around 40 seconds (BLV = 38.25±5.31s and blindfolded sighted = 39.63±6.87s) to search for items before prompting the next stage (reaching for the item). We did not observe any difference in the time when comparing successful/unsuccessful trials and trials with two/three items on the table.

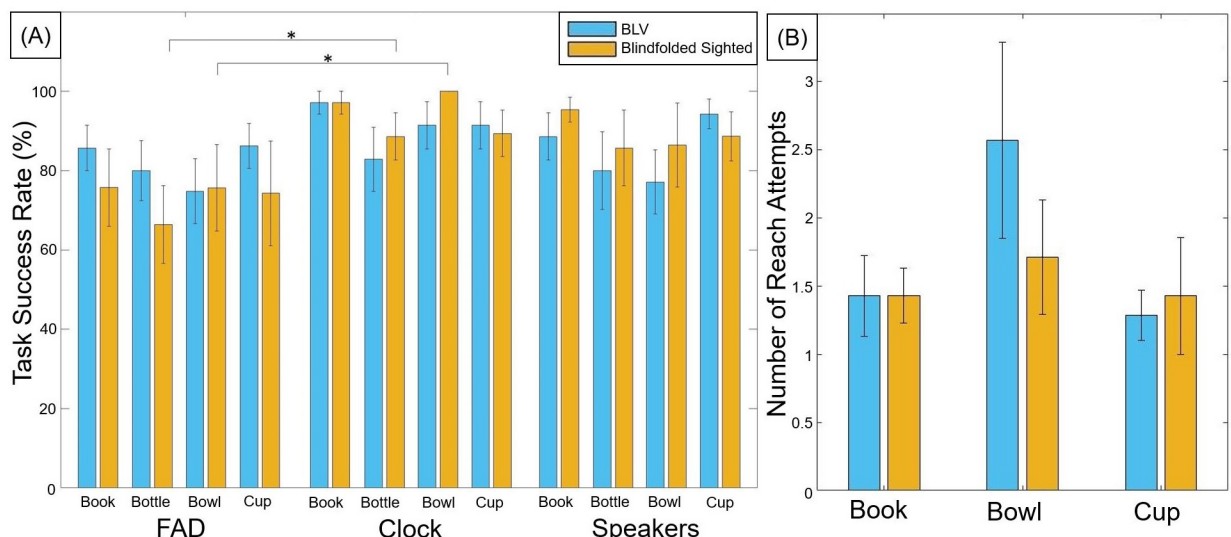

**Fig 8.** (A) The average with standard error bars of the seated task success rate sorted by each item and condition for both the BLV and blindfolded sighted participants. (B) The average with standard error bars of the number of attempts to reach for the target item during the standing task.

### 3.2 Standing task performance

Fig 8B shows the number of reach attempts made by the participants during the standing task. We found that all participants successfully found the target item among the bottles. We recorded the number of reach attempts to gauge the effectiveness of the FAD in differentiating items. We found that participants could accurately differentiate the target items from the bottles with an average of 1.67 attempts to reach the target item (BLV = 1.76±0.75 and blindfolded sighted = 1.52±0.34). We did not observe any difference in average performance between the items. Some participants required more attempts to distinguish the bowl from the bottle.

### 3.3 Hand reaching behaviour

Fig 9 outlines the reaching behaviors of the participant groups. We did not find differences within the BLV participant group for speed, efficiency, and reaching. This result suggests that the participants were consistent in reaching behavior using the FAD compared to the two idealized conditions. In comparison, we observed that the blindfolded sighted participant group was less efficient in reaching the target items. The blindfolded sighted participant group had a significantly higher reaching speed in the FAD condition compared to the Speakers condition ($Z = 2.20$, $p = 0.028$, and $r = 0.83$). However, the trajectory was less efficient ($Z = 2.20$, $p = 0.028$, and $r = 0.83$) and had a longer reaching duration ($Z = 2.03$, $p = 0.043$, and $r = 0.77$) in the FAD condition compared to the Clock condition. This finding suggests that the sighted participant tried to reach the target item with a higher speed but took a much less efficient trajectory.

### 3.4 Head searching behaviour

Fig 10 outlines the results of the participant's head movements. We found that successful trials of both participant groups (Blind, $Z = 2.20$, $p = 0.028$, and $r = 0.83$, and Sighted, $Z = 1.99$,

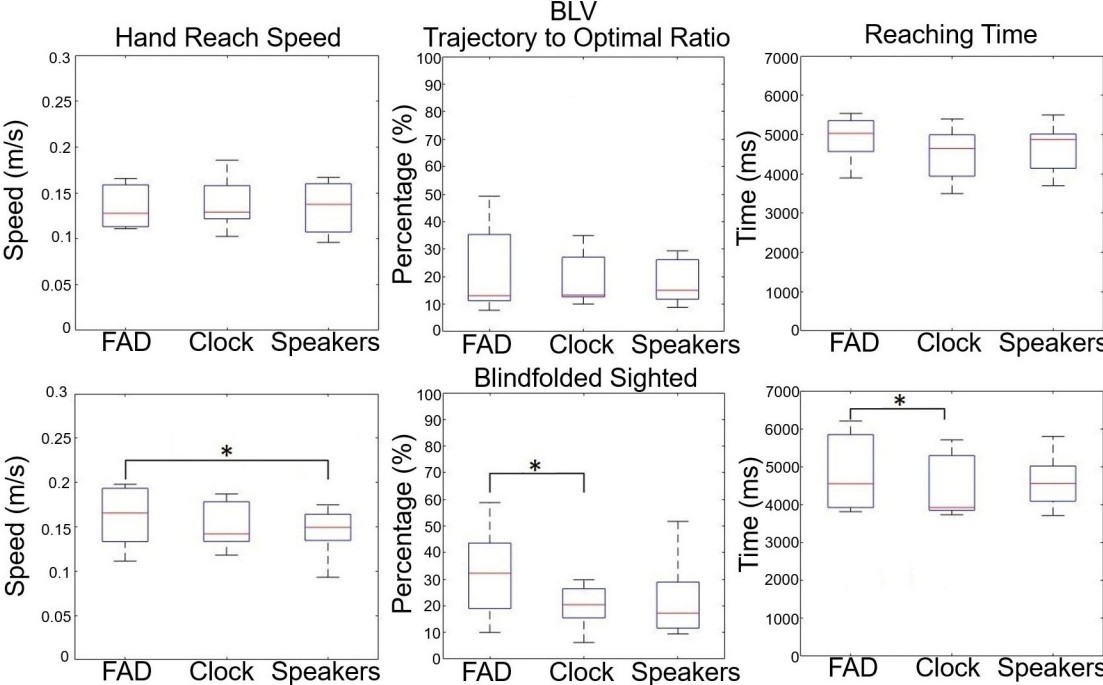

**Fig 9. The box plots show the hand reaching speed, trajectory-to-optimal ratio, and the reaching time for the BLV and blindfolded sighted participant groups.**

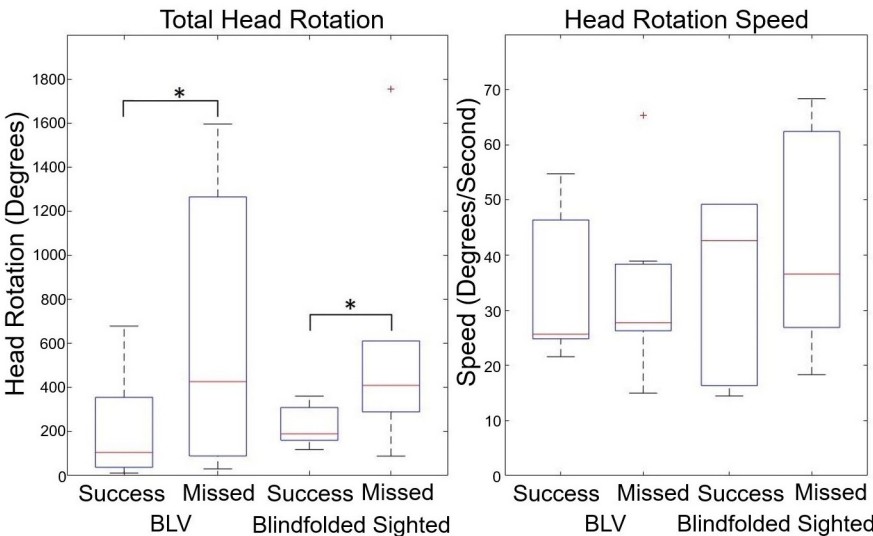

**Fig 10. The box plots for the total head rotation and head rotational speed during the scanning-for-items stage of the FAD condition are shown for the seated task.** It should be noted that the item could be placed directly on either side of the participant; a single head sweep to search the table would be 180˚ rotation.

p = 0.046, and r = 0.75) required significantly less head rotation when compared to the unsuccessful (missed target item) trials. One consideration for this result is that there were far fewer unsuccessful trials (70–90% success rate) which reduced the reliability of this result. Based on observation from the recorded video footage, we observed that participants often over-compensate (BLV = 639.01±647.18 and Sighted = 593.61±597.58) in head rotation (possibly to attain certainty) when the FAD device missed items on the table (e.g., finding one item and missing the second). The successful trials often required fewer sweeps of the table (BLV = 210.14±248.52 and Sighted = 222.12±93.87) to locate and identify the surrounding items. We did not identify patterns for the head rotational speed, with both participant groups exhibiting statistically consistent rotation speeds.

## 3.5 Physiological measurements during the tasks

Table 1 presents a summary of the physiological measurements during the different stages of the study. These measurements provide key insights into the participants' physiological responses (fatigue, workload, stress, and bodily activity). We did not observe any changes in HR, HRV, EDA, and TEMP when comparing the physiological measurements with the study conditions. When comparing the FAD to the Speakers condition, there was a significant change in the BR for the BLV participant group (Z = 2.03, p = 0.043, and r = 0.77) groups. While the BR measurement did show a difference, the lack of change in the other physiological measurements suggests that the participants did not experience a heightened physiological response.

On the other hand, during the standing task, the participants exhibited a significant increase in HR (BLV, Z = 2.37, p = 0.018, and r = 0.89, and Sighted, Z = 2.37, p = 0.018, and r = 0.89), BR (Sighted, Z = 2.03, p = 0.043, and r = 0.77), and TEMP (Sighted, Z = 2.37, p = 0.018, and r = 0.89). This finding suggests that the Sighted participant did experience a heightened physiological response. However, this would be expected as the participant was engaged in full-body movement (in comparison to being seated). These results indicate that it

**Table 1. Summary results are given for HR (beats per minute), HRV (milliseconds), BR (breaths per minute), TEMP (Degrees Celsius), and EDA (microsiemens), the physiology data.** The table outlines each condition's average and standard deviation values and the overall average of the Seated and Standing tasks. The stars outline any statistically significant difference found between the conditions (FAD, Clock, and Speakers) and tasks (Seated and Standing). B.S. refers to the blindfolded sighted participant group. F refers to the FAD condition, Cl refers to the Clock condition, and Sp refers to the Speakers condition.

| Measure | FAD | Cl | Sp | Sig | Seated | Standing | Sig |
|---|---|---|---|---|---|---|---|
| HR-BLV | 67.71±14.59 | 68.12±12.56 | 68.69±14.69 | | 68.18±13.87 | 81.73±17.61 | * |
| HR-B.S. | 74.93±14.37 | 75.07±14.62 | 74.29±15.06 | | 74.76±14.65 | 79.12±15.52 | * |
| HRV-BLV | 47.62±19.13 | 46.49±18.00 | 46.87±18.39 | | 47.00±18.46 | 43.45±19.45 | |
| HRV-B.S | 56.61±18.74 | 57.13±19.43 | 56.50±18.15 | | 56.75±18.76 | 54.82±19.09 | |
| BR-BLV | 14.56±2.78 | 14.62±3.16 | 14.27±2.78 | *F/Sp | 14.49±2.89 | 16.30±2.68 | |
| BR-B.S | 13.96±2.48 | 13.72±2.80 | 14.15±2.60 | | 13.95±2.61 | 15.18±2.42 | * |
| TEMP-BLV | 36.47±0.70 | 36.47±0.70 | 36.48±0.68 | | 36.47±0.69 | 36.78±0.73 | |
| TEMP-B.S | 36.54±0.79 | 36.55±0.79 | 36.57±0.76 | | 36.55±0.78 | 36.94±0.52 | * |
| EDA-BLV | 0.23±0.27 | 0.27±0.42 | 0.26±0.41 | | 0.25±0.36 | 0.80±0.99 | |
| EDA-B.S | 0.31±0.35 | 0.31±0.36 | 0.30±0.38 | | 0.31±0.36 | 0.31±0.68 | |

is unlikely that the participant experienced a significant increase in stress or anxiety levels during the study.

## 3.6 NASA-TLX

Fig 11 outline the NASA-TLX score provided by the participants during each stage. We found that the BLV participant group reported a significant increase in perceived effort during the standing task compared to the training stage ($Z = 2.03$, $p = 0.042$, and $r = 0.77$). We did not find a change in workload levels in the other categories for the BLV participant group. The sighted participant group reported a significant increase in physical demand as the study progressed to the seated ($Z = 2.38$, $p = 0.017$, and $r = 0.90$) and standing task ($Z = 2.03$, $p = 0.042$, and $r = 0.77$). These results suggest that the standing task increases the participant's experience of physical demand and perceived effort. Similar to the physiological results, the NASA-TLX

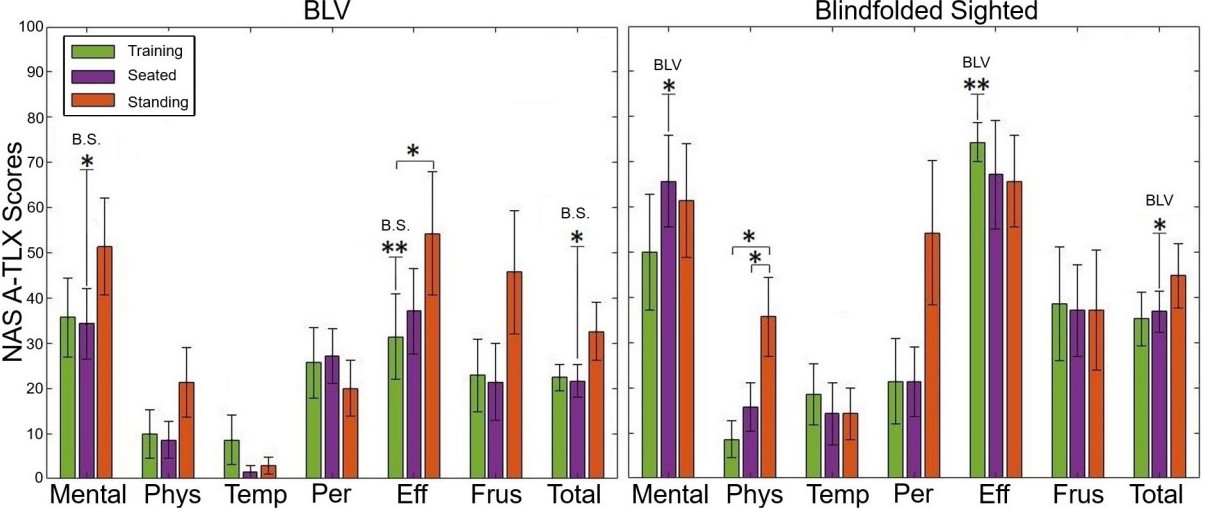

**Fig 11. The average values with standard error bars are shown for each NASA-TLX category and the total (RAW-TLX) workload value for the BLV and Sighted participant groups.** The scoring categories are Mental, Physical (Phys), Temporal (Temp), Performance (Per), Effort (Eff), Frustration (Frus), and Total Raw-TLX score. B.S. refers to the Blindfolded Sighted participants.

results are reasonable, considering the standing task involves a higher level of physical activity. Interestingly, we found that the sighted participant group reported a higher workload score when compared to the BLV participant (Z = 2.05, p = 0.041, and r = 0.77). This finding is likely due to the effect of the blindfold on the sighted participants.

## 4 Discussion

### 4.1 Comparison of FAD with the clock and speakers condition

We compared the FAD to the Clock and Speakers conditions in the seated task. Previous literature demonstrates that the use of clock face instructions [56] or external speakers [57] can effectively aid in localizing objects. Therefore, we consider the Clock and Speakers conditions to be an idealized representation of conventional AT devices. It is important to note that the Clock and Speakers conditions operate with precise prior knowledge of the positions of the items. Thus these conditions have perfect reliability and accuracy in the information presented to the participants. In contrast, in every trial of the FAD, the participant explores an unknown item placement sequence and contends with potential inaccuracies with the machine vision object recognition. Another consideration is the active and passive engagement of the participant. In the FAD trials, the participant actively engages in an item search that requires sensory-motor coupling between the head scanning and auditory feedback while creating a mental spatial map of the item locations. In comparison, the Clock and Speakers conditions were passive searches as participants listened to the auditory feedback and memorized the item locations. Therefore, the Clock and Speakers condition is far less mentally demanding than the FAD condition [58].

In light of these factors, the BLV participants did not exhibit a decrease in overall performance (Fig 7A) between the FAD and the Clock/Speakers conditions. This consistency is also observed in the item detection rate (Fig 7B), the task success rate for each item (Fig 8A), and hand trajectory behavior (Fig 9). This suggests that participants could pick up and use the FAD without a decrease in performance compared to other already familiar methods (clock face instructions and external sound). In contrast, we did observe a significant decrease in performance within the sighted participant group when using the FAD. We believe that the decrease in performance could be attributed to two factors—the less developed blindfolded O&M skills compared to the BLV participants. The latter has far more developed independent O&M skill sets and the unfamiliarity and novelty of the acoustic touch technique.

It is reasonable that blocking a sighted person's vision (using a blindfold) will dramatically reduce their ability to orient and identify objects within their surroundings spatially. As observed in the NASA-TLX results (Fig 1), the sighted participants rated a significantly higher level of mental workload, effort, and total workload when compared to the BLV participants. This finding correlates with the study by Postma et al. [59], which indicates that the sighted blindfolded participants would exhibit worse spatial learning and spatial memory when compared to participants who are blind. The sighted participants showed a significant performance increase over time, but the BLV participant group consistently performed better. We considered this factor and recruited 3 female O&M specialists who have previously received blindfold training. We reasoned that the occupational experience and training would provide a higher baseline level of blindfolded O&M skill and reduce the shock/inexperience of performing tasks without sight. Interestingly, the lack of difference in the overall task success rate between the O&M specialists (64.80±26.15%) and the non-specialist sighted participants (79.17±11.90%) suggests that the decrease in performance is more likely due to the unfamiliarity of using acoustic touch or the FAD. It could be argued that the BLV participants were also unfamiliar with the acoustic touch. However, many smart glasses [31, 39] and other handheld [60] AT

devices use similar concepts of motor-sensory coupling and sensory augmentation. Therefore, it is likely that the BLV participant group had some general familiarity with AT devices that contributed to learning and using the acoustic touch/FAD. Overall, the results provide a strong indication that the FAD device, when given to a user who is blind, can be used with familiarity and a similar level of performance to other AT devices.

**4.1.1 FAD: Head scanning, item sonification, and spatial mapping.** From the total head rotation results, it is clear that in the successful trials, both participant groups required significantly less head rotation compared to the unsuccessful trials. This result seems counter-intuitive; if a person scans the environment more, it would be reasonable to have more information. However, we believe the data suggests a crucial factor in the reliability of the FAD's object recognition. If the unsuccessful trials were due to issues with the acoustic touch technique, then it would be expected to be observed a decrease in task success rate compared to the Clock and Speaker conditions, which involve no head scanning. It could be reasoned that the participants may rely on an ineffective head scanning method, such as scanning too fast. However, the head rotation speed did not yield any significant difference between successful and unsuccessful trials. This suggests that the participants were consistent with their head scanning speed. We believe the unsuccessful trials were due to errors in object recognition rather than the acoustic touch. We argue that the increased head rotation/sweeps may have introduced a higher risk of incorrect/misinformation from the object recognition, which led to some unsuccessful trials. An essential future consideration will be implementing more robust object recognition methods that can compensate for the head scanning motion and the quality of the head-mounted cameras.

We evaluated the item detection rate (Fig 7) of the seated task and the standing performance to understand better the reliability of using auditory icons (instead of automated text, i.e., Clock condition) in item identification. The standing task indicates the participants' ability to detect specific items within a cluttered environment. All participants successfully identified the book, bowl, and cup within an environment filled with bottles. When observing the seated task success rate by items, we found that the sighted participants had a lower success rate for the bottle and bowl sounds when compared to the automated speech of the Clock condition. The bottle and bowl sounds (glass scraping/collision sounds) were confusing to the participants. One interesting observation is that participants could correct confused items in trials containing one glass item (i.e., book + bottle/cup/bowl) as they could deduce the item identity even though it was incorrectly recognized. Based on this assessment, we can conclude that auditory icons can convey the identity of an item. However, careful consideration should be given to selecting the auditory icon to ensure a unique item identity.

One drawback of the clock face instructions is the lack of precise spatial information. Each clock face zone represents roughly 36° (180° across 5 clock regions) of the area and does not provide any distance information. The FAD and Speakers can covey more precise directional information to the participant. From the observation and development, the FAD allows directional information through the variable repetition of the sound (higher repetition rate when closer to the center of FOV), and the distance can be gauged using vertical head scanning. In contrast, the Speakers condition requires apriori knowledge that is not translatable to real-world use. The task performance and reaching accuracy results indicate that the acoustic touch technique can convey accurate spatial information. In summary, the acoustic touch paradigm can successfully enable the recognition and localization of an item. However, future iterations of wearable devices should emphasize the reliability of machine object recognition and carefully design auditory icons for objects.

**4.1.2 Cognitive workload and physiological stress.** When observing the NASA-TLX scores (Fig 1), we found that the sighted participant group experienced a higher cognitive

workload than the BLV participant group. These findings are unsurprising, as the sighted participants are likely adjusting and adapting to the blindfolded. The BLV participants, on average, rated an increased perceived effort over the study stages. This result is likely due to the standing task stage's more active nature than the training, which was a relatively relaxed task (participants were given unlimited time/trials to learn the protocol). Overall, the NASA-TLX scores suggest that the prolonged use of the FAD did not significantly increase the participants' workload level.

When we compared the physiological measures (Table 1) between the seated task conditions, we did not observe a significant increase in physiological activity. The BR is the exception to this observation in that a significant increase was observed for the BLV participant group between the FAD and Speakers conditions. While an increase in BR could be interpreted as a mild hyperventilation [61] behavior indicating an increase in anxiety, it can more likely be attributed (due to lack of change within the other metrics) to increased bodily movement [62]. The significant increase in HR, BR, and TEMP between the seated and standing tasks for the blindfolded sighted participant group suggest an increase in acute stress levels [63]. The increase in HR can likely be attributed to the uncertainty of moving into an unfamiliar environment without their mobility aid. Additionally, the blindfolded sighted participants would be contending with the uncertainty of walking with a blindfold resulting in increased BR and TEMP. If given more time or repeated sessions, these users could become more familiar with the device and experience less stress during the standing task.

## 4.2 Limitations

Multiple similar previous works [64–67] have explored using the YOLOv5 (or similar) deep learning model on a smartphone for the computer vision of smart glasses. These studies have found many successes of the technology. However, we observed two unmentioned technical limitations that may hinder the device's future usability. Firstly, there is an issue regarding the robustness of object recognition. As outlined in the discussion, the reliability of object recognition is crucial to the success of the FAD. For the FAD, we limited the object recognition to four object classes with specific easily recognized items. This choice was made to ensure a reliable level of performance during the user study. In its current state, the FAD cannot be feasibly relied upon in a real-world situation unless the robustness of the object recognition is vastly improved. Another limitation is the computation load of object recognition on the smartphone. The OPPO smartphone is a high-performance phone [43], yet the demands of real-time head tracking, object recognition, and item sonification caused significant heating and power consumption issues. Future iterations and development of AT smart glasses must resolve these issues to provide a usable solution for the end-user.

## 4.3 Conclusion

We explored the concept of using an acoustic touch paradigm as a sensory augmentation method to aid people who are BLV in recognizing and picking up household objects. We presented an outline of the described development of the FAD study protocol, which was used to investigate the efficacy and usability of the FAD using an end-user study with BLV and blindfolded sighted participants. Our results from the questionnaire, user performance, behavior (hand and head), and physiological measurements indicated that the acoustic touch paradigm is an effective method for presenting spatial and semantic information about nearby items. Future iterations of acoustic touch paradigms should improve the object recognition's reliability, refine the system's usability, and address the device's technical limitations concerning power consumption and heating. Overall, the study suggests that the acoustic touch paradigm

could be further developed, improved, and incorporated into AT intended to support people with BLV to access their environment more efficiently and effectively.

## Supporting information

**S1 Questionnaire. Questionnaire sheet.** The questionnaire sheet that was used during the experiment.
(PDF)

**S1 Video. Video introduction to acoustic touch.** A video explaining the concept of acoustic touch and the functionality of the FAD device.
(MP4)

**S2 Video. Video of seated task.** A video explaining the seated task of the experiment. The video outlines the conditions and the stages of each trial.
(MP4)

**S3 Video. Video of standing task.** A video explaining the standing task of the experiment. The video outlines the conditions and the stages of each trial.
(MP4)

**S1 Text. Captions for the video introduction to acoustic touch.** The subtitle file for the video explaining the concept of acoustic touch and the functionality of the FAD device.
(SRT)

**S2 Text. Captions for the video of seated task.** The subtitle file for the video explaining the seated task of the experiment.
(SRT)

**S3 Text. Captions for the video of standing task.** The subtitle file for the video explaining the standing task of the experiment.
(SRT)

## Acknowledgments

We want to thank R. Yearsley, M. Harrison, N. Jeffers, and G. Dickins from ARIA research for providing constructive feedback and support during the development of the study protocol and the FAD device. We would also like to thank our participants for their time and feedback throughout the study.

## Author Contributions

**Conceptualization:** Howe Yuan Zhu, Shayikh Nadim Hossain, Avinash K. Singh.

**Data curation:** Howe Yuan Zhu.

**Formal analysis:** Howe Yuan Zhu, Minh Tran Duc Nguyen.

**Funding acquisition:** Craig Jin, Avinash K. Singh, Chin-Teng Lin.

**Investigation:** Howe Yuan Zhu, Shayikh Nadim Hossain, Craig Jin, Avinash K. Singh, Lil Deverell, Vincent Nguyen, Julee-anne Renee Bell, Chin-Teng Lin.

**Methodology:** Howe Yuan Zhu, Shayikh Nadim Hossain, Craig Jin, Minh Tran Duc Nguyen, Lil Deverell, Vincent Nguyen, Felicity S. Gates, Julee-anne Renee Bell.

**Project administration:** Howe Yuan Zhu, Lil Deverell, Felicity S. Gates.

**Resources:** Howe Yuan Zhu, Ibai Gorordo Fernandez, Marx Vergel Melencio.

**Software:** Howe Yuan Zhu, Ibai Gorordo Fernandez, Marx Vergel Melencio.

**Supervision:** Howe Yuan Zhu, Craig Jin, Avinash K. Singh, Chin-Teng Lin.

**Validation:** Howe Yuan Zhu.

**Visualization:** Howe Yuan Zhu.

**Writing – original draft:** Howe Yuan Zhu.

**Writing – review & editing:** Howe Yuan Zhu, Shayikh Nadim Hossain, Craig Jin, Minh Tran Duc Nguyen, Lil Deverell, Vincent Nguyen.

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
