## [Decision Letter · Decision Letter 0]

9 Jun 2023

PONE-D-23-03381An Investigation into the Effectiveness of using Acoustic Touch to Assist People who are BlindPLOS ONE

Dear Dr. Zhu,

Thank you for submitting your manuscript to PLOS ONE. After careful consideration, we feel that it has merit but does not fully meet PLOS ONE’s publication criteria as it currently stands. Therefore, we invite you to submit a revised version of the manuscript that addresses the points raised during the review process. Please find the reviewer comments below for your consideration.

We look forward to receiving your revised manuscript.

Kind regards,

Iftikhar Ahmed Khan

Academic Editor

PLOS ONE

Journal Requirements:

a) Did participants provide their written or verbal informed consent to participate in this study?

“This work was supported by the Australian Cooperative Research Centres Projects (CRC-P) Round 11 CRCPXI000007, the ARIA research, the University of Technology Sydney, and the University of Sydney. Received by C.J, V.N and C.L.”

Reviewers' comments:

Reviewer's Responses to Questions

**Comments to the Author**

1. Is the manuscript technically sound, and do the data support the conclusions?

Reviewer #1: Yes

Reviewer #2: Yes

2. Has the statistical analysis been performed appropriately and rigorously? 

Reviewer #1: Yes

Reviewer #2: Yes

3. Have the authors made all data underlying the findings in their manuscript fully available?

Reviewer #1: Yes

Reviewer #2: Yes

4. Is the manuscript presented in an intelligible fashion and written in standard English?

Reviewer #1: Yes

Reviewer #2: Yes

5. Review Comments to the Author

Reviewer #1: The paper on acoustic touch to provide a wearable solution is well written with necessity information on Foveated Audio Device for assisting people who are blind (BLV) in finding objects.

Intensive study has been done with nearly 60 Papers.

Physiological Behaviour of person were also addressed for Analysis.

The figure placement is done correctly.

Suggestions:

1. Performance metrics of different TASKs can be explained in detail with mathematics functions.

2. Significant details on Kolmogorov-Smirnov test, Mann-Whitney U-test Wilcoxon rank sum test need to be provided which aids Statistical Analysis.

3. Justifying Table 1 (Physiological Behaviour w.r.t FAD outcomes)

4. Comparison of FAD Outcomes with necessary Tabulation.

Reviewer #2: This paper explored the potential of a new technique known as ”acoustic touch” to provide a wearable solution for assisting people who are blind in finding objects. Authors developed a wearable Foveated Audio Device to study the efficacy and usability of using acoustic touch to search, memorize, and reach items. I have the following concerns:

1. The introduction is written well. However, the related works aren’t enough to get the current state and gaps of this research field, and AI based methods in the last five years also must be added as Section 2.

2. It would be nice if authors can provide a table to provide the methods used in previous studies and the drawbacks and advantages of similar studies mentioned in the related work.

3. Please improve quality of all Figures.

4. Please add figure to demonstrate overall framework of the proposed system in Section 2.

5. How do you calculate distance between camera and objects? Please add detail explanation for this process.

6. Why is YOLOv5 model selected for object recognition? Which YOLOv5 module is used? There are multiple YOLOv5 modules with different size. Please clarify this selection.

7. Comparing this work with other similar works is necessary. Thus, please add tables or figures to prove this work's advantages over other similar works.

8. In general, Section 3 is must be modified with more evaluations and comparisons with other popular methods. Is it possible to compare this work with object recognition based works? It could be great contribution for this research area.

9. How does this method work with higher resolution images?

10. The table of all modules, components and parameters must be included in the modification of this work and the reference and setting support of all parameters and the datasheets must be labelled to prove the physical implementability and repeatability.

11. The performance metrics in computation complexity, processing delay, power consumption cost and the relevant gains for the proposed scheme must be included.

12. Please prove “4.1.2 Cognitive Workload and Physiological Stress” section with table or figures.

13. However, it is not clear how their proposal will be efficient in real -time implementation.

14. Please check the style and format of the references.

15. To improve the Related Work and Introduction sections authors are recommended to review this highly related research work:

a. Smart glass system using deep learning for the blind and visually impaired

6. PLOS authors have the option to publish the peer review history of their article (what does this mean?). If published, this will include your full peer review and any attached files.

Reviewer #1: No

Reviewer #2: **Yes: **Mukhriddin Mukhiddinov

---

## [Author Response · Author response to Decision Letter 0]

13 Jul 2023

Thank you to the editor and reviewers for your time and feedback in improving our manuscript. We have reviewed the reviews and revised our paper according to the feedback.

Editor Feedback:

Thank you for your time and effort in organizing the reviews and aiding in reviewing the manuscript.

Response: We've ensured that the current submission follows the PLOS One style. The manuscript is generated from the Latex template provided at: https://journals.plos.org/plosone/s/latex#loc-how-to-submit

We used version 3.6; if this version is incorrect or there are any specific issues with the manuscript, please let us know, and we can rectify the issue.

a) Did participants provide their written or verbal informed consent to participate in this study?

Response: Our participants who were BVI provided verbal consent during the pre-experiment screening interview. Our O&M specialist would verbally read the consent form, answer questions, and ask for formal consent. If the participants consented to participate, they would be noted on the interview form as consent is given. 

The sighted participants provided written consent on the form directly. 

Response: i) This was primarily for the convenience of the BVI participants. Consent was collected during the screening interview with our O&M specialist. We acknowledge that braille and screen readers were options. We found that reading the consent form directly to the participants can avoid confusion/miscommunication and allow the participants to ask questions. Participants were formally asked if they consented to each item (de-identified recorded data and video/photograph/audio). The O&M specialist documented the consent of the participants with BVI on the intake form, as the participant verbally relayed the details. 

ii) The consent was documented on our interview intake form. The O&M specialist would fill out the form during the screening interview for record keeping. The sighted participants would directly fill out the form. 

iii) We can confirm that our human research ethics committee received and approved all procedures and protocols for this study. 

We have revised the paper to reflect the outlined points.

"This work was supported by the Australian Cooperative Research Centres Projects (CRC-P) Round 11 CRCPXI000007, the ARIA research, the University of Technology Sydney, and the University of Sydney. Received by C.J, V.N and C.L."

Please provide an amended statement that declares *all* the funding or sources of support (whether external or internal to your organization) received during this study, as detailed online in our guide for authors at http://journals.plos.org/plosone/s/submit-now. Please also include the statement "There was no additional external funding received for this study." in your updated Funding Statement. Please include your amended Funding Statement within your cover letter. We will change the online submission form on your behalf.

Response: The CRC-P funding is the sole funding for this particular study. We have amended the statement as suggested:

This work was supported by the Australian Cooperative Research Centres Projects (CRC-P) Round 11 CRCPXI000007, the ARIA research, the University of Technology Sydney, and the University of Sydney. Received by C.J, V.N and C.L. There was no additional external funding received for this study.

4. We note that Figure 1, 2,5 & 6 includes an image of a [patient / participant / in the study]. 

As per the PLOS ONE policy (http://journals.plos.org/plosone/s/submission-guidelines#loc-human-subjects-research) on papers that include identifying, or potentially identifying, information, the individual(s) or parent(s)/guardian(s) must be informed of the terms of the PLOS open-access (CC-BY) license and provide specific permission for publication of these details under the terms of this license. Please download the Consent Form for Publication in a PLOS Journal (http://journals.plos.org/plosone/s/file?id=8ce6/plos-consent-form-english.pdf). The signed consent form should not be submitted with the manuscript, but should be securely filed in the individual's case notes. Please amend the methods section and ethics statement of the manuscript to explicitly state that the patient/participant has provided consent for publication: "The individual in this manuscript has given written informed consent (as outlined in PLOS consent form) to publish these case details". 

Response: All participants shown provided consent for photography and video recording during the experiment. The consent outlined that the footage would be de-identified/censored (blurred face) and used in publications/presentations. The ethics committee approved this form and procedure. We have revised the paper as recommended.

Reviewer 1 

The paper on acoustic touch to provide a wearable solution is well written with necessity information on

Foveated Audio Device for assisting people who are blind (BLV) in finding objects.

Intensive study has been done with nearly 60 Papers.

Physiological Behaviour of person were also addressed for Analysis.

The figure placement is done correctly.

Thank you for the comments and suggestions for improving the manuscript; we highly appreciate the feedback and hope that we have adequately addressed the comments.

Suggestions:

1. Performance metrics of different TASKs can be explained in detail with mathematics functions.

Thank you for the suggestion; we have revised the methodology with mathematical equations to better clarify the performance metric calculation.

2. Significant details on Kolmogorov-Smirnov test, Mann-Whitney U-test Wilcoxon rank sum test need to be provided which aids Statistical Analysis.

Response: Thank you for pointing out the missing detail. We have revised the data analysis to include the p-value for the normality test. All the metrics significantly differed from the CDF, indicating a non-normal distribution. We have ensured the values (Z, p, and r values) are reported in the results. Please let us know if any other specific indicators need to be included.

3. Justifying Table 1 (Physiological Behaviour w.r.t FAD outcomes)

Response: Thank you for the suggestion; we've revised the results section with a clearer justification for Table 1.

4. Comparison of FAD Outcomes with necessary Tabulation.

Response: We have ensured that the discussion compares the functional and human factors of the FAD. We compare the wearable binaural auditory icons (FAD) against using speech and speaker generate audio (external sound). The FAD is merely a tool/platform to test wearable binaural auditory icons. In terms of outcomes, the discussion explores functional performance (comparison between conditions), usability (body kinematics and ergonomics), and human factors (cognition, physiology, and emotional response). We are happy to improve the clarity if specific aspects need to be clarified.

Reviewer 2

This paper explored the potential of a new technique known as" acoustic touch" to provide a wearable solution for assisting people who are blind in finding objects. Authors developed a wearable Foveated Audio Device to study the efficacy and usability of using acoustic touch to search, memorize, and reach items. I have the following concerns:

Response: Thank you for spending the time to review and provide feedback for this paper. 

This work primarily investigates the acoustic touch technique as an audio presentation technique. This technique uses head-orientated auditory icons on the binaural audio device, which differs from computer-generated speech and monotone (non-spatialized) sounds used in conventional assistive devices. Likewise, the novelty of this work is the investigation of audio presentation methods (Acoustic Touch vs Speech) and rendering solutions for the audio presentation (the FAD wearable binaural vs external speaker). 

We apologize for any confusion within the paper; from the feedback, there is a misunderstanding that this is a systems paper or an ML device/technology study. This paper is not intended to investigate the ML model's performance or the FAD as a real-world implementation device. We are investigating the efficacy and usability of the acoustic presentation technique through a user study. The main contributions of this type of investigation and paper are to the area of sensory augmentation and psychoacoustic. 

I agree that the points raised are valid and important for the ML field. However, it is outside the scope of this paper. Many members of the authorship are actively developing better object recognition for wearable glasses. This paper is not intended to innovate on the ML/real-time classification techniques. The initial stages of the FAD did not include object recognition (purely motion tracking driven); we only included the basic object recognition model to enable testing the acoustic touch technique on the glasses. 

Again, we apologize for the confusion in the paper; we have addressed the AI-related comments to the best we can without diluting the scope of the paper. We hope this clarifies the scope and goals of the paper.

1. The introduction is written well. However, the related works aren't enough to get the current state and gaps of this research field, and AI based methods in the last five years also must be added as Section 2.

Response: Thank you for raising this point; we agree there is a challenge and research gap for head-worn real-time object recognition. However, this paper is not intended to explore different AI-based methods. A comprehensive survey of ML-based methods would likely cause more confusion for our intended audience (assistive technology, sensory augmentation, and psychoacoustics) as the study does not offer innovation or metrics in the ML research field. The introduction focuses on sensory augmentation techniques and assistive technologies because the study evaluates the performance of auditory sensory augmentation for auditory icons vs speech. We have revised the introduction to highlight better the core areas of the paper and the research gap within the field. 

2. It would be nice if authors can provide a table to provide the methods used in previous studies and the drawbacks and advantages of similar studies mentioned in the related work.

Response: As previously highlighted, the core comparison is the acoustic touch technique (a relatively new concept) to computer-synthesized speech (all consumer products utilize this). We have revised the introduction to highlight better the difference between acoustic touch and computer-synthesized speech. 

3. Please improve quality of all Figures.

Response: Based on the feedback, we have double-checked the resolution and formatting of the figures. The current figure format and resolution are based on the figure guidelines by PLOSONE (https://journals.plos.org/plosone/s/figures). Some quality may have been lost when converting the figure to the PLOSONE guideline. 

Another note is that the figures generated on the PLOSOne submission PDF document are compressed to a lower resolution. Downloading the figure will provide the image at full resolution. 

If the quality issue on all figures differs entirely from the above, let us know about the specific issue, and we'll happily address it.

4. Please add figure to demonstrate overall framework of the proposed system in Section 2.

Response: Thank you for the suggestion; please note that Figure 4 is a block diagram that outlines the overall framework and components of this paper/study. As previously mentioned, this paper is not intended to present FAD as a system but as a testing platform for the acoustic touch technique. We can revise or add a figure if the reviewer has a particular figure in mind. 

5. How do you calculate distance between camera and objects? Please add detail explanation for this process.

Response: We have revised the methodology (2.1.2) with a detailed explanation of the calculation for distance.

6. Why is YOLOv5 model selected for object recognition? Which YOLOv5 module is used? There are multiple YOLOv5 modules with different size. Please clarify this selection.

Response: We have revised the methodology (2.1.2) with a detailed explanation of the specific YOLOv5 model (m) and the other tested models. 

7. Comparing this work with other similar works is necessary. Thus, please add tables or figures to prove this work's advantages over other similar works.

Response: Thank you for the suggestion; we have revised the introduction to highlight the advantages of this work better. 

8. In general, Section 3 is must be modified with more evaluations and comparisons with other popular methods. Is it possible to compare this work with object recognition based works? It could be great contribution for this research area.

Response: Thank you for the suggestion. Unfortunately, an extensive review of object recognition-based works is not within the scope of the paper. As previously highlighted, object recognition is a small component of the paper. The core component is exploring the presentation of audio cues, which is valuable to the sensory augmentation field. In the introduction, we have highlighted several works and commercial assistive devices that use synthesized speech. These works align with the intended scope of the work. 

9. How does this method work with higher resolution images?

Response: The input size was chosen to balance accuracy, inference time, and app stability. At higher resolutions, we found that the inference took longer, and the app was less stable (due to the integration of audio and head tracking). However, this was not robustly tested as the object recognition was only used to enable head-oriented auditory icons for the items for the study. 

10. The table of all modules, components and parameters must be included in the modification of this work and the reference and setting support of all parameters and the datasheets must be labelled to prove the physical implementability and repeatability.

Response: We have revised Section 2.1, and Figure 4 outlines the necessary components for reproducing the work. 

11. The performance metrics in computation complexity, processing delay, power consumption cost and the relevant gains for the proposed scheme must be included.

Response: Thank you for the suggestion. We have revised the methodology (2.1.2) with the inference time and power consumption of the app version used for this study.

12. Please prove "4.1.2 Cognitive Workload and Physiological Stress" section with table or figures.

Response: Thank you for highlighting this issue. We have revised section 4.1.2 to include the references to the table/figures derived from the discussion.

13. However, it is not clear how their proposal will be efficient in real -time implementation.

Response: Apologies for the misunderstanding in the paper. The FAD's main function is to enable the evaluation of the acoustic touch technique. The device is not intended to be an in-the-wild technology. As highlighted in the paper, the model was limited to 4 items and used in a controlled environment. The only design condition is for a device that can work reliably within the study design to evaluate the research questions/goals of the paper. The primary contribution of this paper is to explore the auditory presentation, not to generate a real-time/real-world AT device. 

For these reasons, we explicitly state in the introduction, "To evaluate our hypothesis, we developed a Foveated Audio Device (FAD) to investigate the use of acoustic touch in AT". The sole purpose of the FAD is to evaluate the acoustic touch technique. 

14. Please check the style and format of the references.

Response: Thank you for pointing out this issue; we have fixed the issues with the references. The references are generated through Bibtex and formatted through the PLOSONE template. Please let us know if there are any specific issues that we missed.

15. To improve the Related Work and Introduction sections authors are recommended to review this highly related research work:

a. Smart glass system using deep learning for the blind and visually impaired

Response: Thank you for suggesting this relevant paper; we have included it in the paper.

---

## [Decision Letter · Decision Letter 1]

9 Aug 2023

An Investigation into the Effectiveness of using Acoustic Touch to Assist People who are Blind

PONE-D-23-03381R1

Dear Dr. Zhu,

We’re pleased to inform you that your manuscript has been judged scientifically suitable for publication and will be formally accepted for publication once it meets all outstanding technical requirements.

Kind regards,

Iftikhar Ahmed Khan

Academic Editor

PLOS ONE

Additional Editor Comments (optional):

Reviewers' comments:

Reviewer's Responses to Questions

**Comments to the Author**

1. If the authors have adequately addressed your comments raised in a previous round of review and you feel that this manuscript is now acceptable for publication, you may indicate that here to bypass the “Comments to the Author” section, enter your conflict of interest statement in the “Confidential to Editor” section, and submit your "Accept" recommendation.

Reviewer #1: All comments have been addressed

Reviewer #2: All comments have been addressed

2. Is the manuscript technically sound, and do the data support the conclusions?

Reviewer #1: Yes

Reviewer #2: Yes

3. Has the statistical analysis been performed appropriately and rigorously? 

Reviewer #1: Yes

Reviewer #2: Yes

4. Have the authors made all data underlying the findings in their manuscript fully available?

Reviewer #1: Yes

Reviewer #2: Yes

5. Is the manuscript presented in an intelligible fashion and written in standard English?

Reviewer #1: Yes

Reviewer #2: Yes

6. Review Comments to the Author

Reviewer #1: The reviewer suggestions were addressed.

Correction in figure citation, the citation of figure 4 in line 130 is cited before figure 3 which is mentioned in line 142. Do incorporate the changes before final proof.

Reviewer #2: The author has thoughtfully addressed my comments and made necessary modifications to the manuscript. The responses provided align with scientific rigor and enhance the overall quality of the research. The manuscript now reflects a cohesive and scientifically sound presentation of the study's objectives, methodology, and findings.

7. PLOS authors have the option to publish the peer review history of their article (what does this mean?). If published, this will include your full peer review and any attached files.

Reviewer #1: No

Reviewer #2: **Yes: **Mukhriddin Mukhiddinov

---

## [Editor Report · Acceptance letter]

27 Sep 2023

PONE-D-23-03381R1 

An Investigation into the Effectiveness of using Acoustic Touch to Assist People who are Blind 

Dear Dr. Zhu:

I'm pleased to inform you that your manuscript has been deemed suitable for publication in PLOS ONE. Congratulations! Your manuscript is now with our production department. 

Kind regards, 

on behalf of

Dr. Iftikhar Ahmed Khan 

Academic Editor

PLOS ONE